# SMURF2 phosphorylation at Thr249 modifies glioma stemness and tumorigenicity by regulating TGF-β receptor stability

Manami Hiraiwa[1,8], Kazuya Fukasawa[1,8], Takashi Iezaki [1,8✉], Hemragul Sabit[2], Tetsuhiro Horie[1], Kazuya Tokumura[1], Sayuki Iwahashi[1], Misato Murata[1], Masaki Kobayashi[1], Akane Suzuki[1], Gyujin Park[1], Katsuyuki Kaneda [3], Tomoki Todo [4], Atsushi Hirao[5,6], Mitsutoshi Nakada[2] & Eiichi Hinoi [1,7✉]

Glioma stem cells (GSCs) contribute to the pathogenesis of glioblastoma, the most malignant form of glioma. The implication and underlying mechanisms of SMAD specific E3 ubiquitin protein ligase 2 (SMURF2) on the GSC phenotypes remain unknown. We previously demonstrated that SMURF2 phosphorylation at Thr[249] (SMURF2[Thr249]) activates its E3 ubiquitin ligase activity. Here, we demonstrate that SMURF2[Thr249] phosphorylation plays an essential role in maintaining GSC stemness and tumorigenicity. *SMURF2* silencing augmented the self-renewal potential and tumorigenicity of patient-derived GSCs. The SMURF2[Thr249] phosphorylation level was low in human glioblastoma pathology specimens. Introduction of the *SMURF2*[T249A] mutant resulted in increased stemness and tumorigenicity of GSCs, recapitulating the *SMURF2* silencing. Moreover, the inactivation of SMURF2[Thr249] phosphorylation increases TGF-β receptor (TGFBR) protein stability. Indeed, *TGFBR1* knockdown markedly counteracted the GSC phenotypes by *SMURF2*[T249A] mutant. These findings highlight the importance of SMURF2[Thr249] phosphorylation in maintaining GSC phenotypes, thereby demonstrating a potential target for GSC-directed therapy.

[1] Department of Bioactive Molecules, Pharmacology, Gifu Pharmaceutical University, Gifu 501-1196, Japan. [2] Department of Neurosurgery, Graduate School of Medical Science, Kanazawa University, Kanazawa, Ishikawa, Japan. [3] Laboratory of Molecular Pharmacology, Division of Pharmaceutical Sciences, Kanazawa University Graduate School, Kanazawa, Ishikawa 920-1192, Japan. [4] Division of Innovative Cancer Therapy, Institute of Medical Science, The University of Tokyo, Tokyo, Japan. [5] Cancer and Stem Cell Research Program, Division of Molecular Genetics, Cancer Research Institute, Kanazawa University, Kanazawa, Ishikawa, Japan. [6] WPI Nano Life Science Institute (WPI-Nano LSI), Kanazawa University, Kanazawa, Ishikawa, Japan. [7] United Graduate School of Drug Discovery and Medical Information Sciences, Gifu University, Gifu, Japan. [8]These authors contributed equally: Manami Hiraiwa, Kazuya Fukasawa, Takashi Iezaki. ✉email: iezaki-ta@gifu-pu.ac.jp; hinoi-e@gifu-pu.ac.jp

S MAD specific E3 ubiquitin protein ligase 2 (SMURF2) is the E3 ubiquitin ligase responsible for specifying the substrates for ubiquitination and degradation by proteasomes[1,2]. Accumulating evidence indicates SMURF2 regulates a wide array of physiological processes, including cell proliferation, invasion, self-renewal, and migration, through its regulation of a variety of signaling pathways[3–5]. The E3 ubiquitin ligase activity of SMURF2 is regulated at the post-transcriptional level through SUMOylation, methylation, and phosphorylation[6–8], as well as at the transcriptional level[9]. We recently demonstrated that the phosphorylation of SMURF2 at Thr[249] (SMURF2[Thr249]) by extracellular signal-regulated kinase 5 (ERK5) plays an essential role in maintaining the stemness of mesenchymal stem cells (MSCs), which contributes to skeletogenesis[10]. Mechanistically, SMURF2[Thr249] phosphorylation activates its E3 ubiquitin ligase activity, which modifies the stability of SMAD proteins, which in turn transcriptionally activate the expression of SOX9, the principal transcription factor of skeletogenesis in MSCs.

Gliomas, which represent ~80% of all primary malignant brain tumors in humans, can be categorized into four grades according to the World Health Organization (WHO) classification criteria: grade I, grade II, grade III, and grade IV (glioblastoma, GBM)[11,12]. GBM, the most malignant form of glioma, is one of the most aggressive and deadly types of cancer. Patients with GBM have a very poor prognosis, with a 5-year survival rate of only 5.1%[13,14]. Glioma stem cells (GSCs), also known as glioma-initiating cells, are a subpopulation of tumor cells that exhibit stem cell-like capacities such as self-renewal and tumor-initiating capacities[15–17]. Recent studies have determined that GSCs contribute to high rates of therapeutic resistance and rapid recurrence[18,19], cancer invasion, immune evasion, tumor angiogenesis, and the recruitment of tumor-associated macrophages, which indicates that targeting GSCs is an efficacious strategy for improving GBM treatment[20–22].

Transforming growth factor-β (TGF-β) signaling, which is tightly regulated through protein ubiquitination[23,24], has been shown to play a crucial role in maintaining the stemness and tumorigenicity of GSCs through several pathways including the SMAD-SOX4-SOX2 axis and the SMAD-LIF-JAK-STAT pathway[25,26]. SMAD7 acts as a scaffold protein to recruit SMURF2 to the TGF-β receptor (TGFBR) complex to facilitate its ubiquitination[27]. This leads to the proteasome-mediated degradation of TGFBRs and the attenuation of TGF-β signaling. Ubiquitin-specific peptidase 15 (USP15), a deubiquitinating enzyme, binds to the SMAD7-SMURF2 complex and deubiquitinates and stabilizes TGFBR1, resulting in enhanced TGF-β signaling[28,29]. The balance between USP15 and SMURF2 activities determines the activity of TGF-β signaling and subsequent oncogenesis in GBM. Indeed, a deficiency in USP15 decreases the oncogenic capacity of GSCs due to the repression of TGF-β signaling[28]; conversely, USP15 amplification confers poor prognosis in individuals with GBM[30]. However, although SMURF2 should be assumed to play an opposite role from that of USP15, no reports have yet directly addressed the implication and underlying mechanisms of SMURF2 on the GSC phenotypes and subsequent glioma pathogenesis both in vivo and in vitro.

In this study, we reveal that SMURF2 silencing by shRNA resulted in an augmentation of the self-renewal potential and tumorigenicity of GSCs. The SMURF2[Thr249] phosphorylation level was downregulated in GBM patients, regardless of the lack of marked changes in its mRNA and protein levels. In addition, the SMURF2[Thr249] phosphorylation level was lower in GSCs than that in differentiated glioma cells. The inactivation of SMURF2[Thr249] phosphorylation by a non-phosphorylatable mutant (SMURF2[T249A] mutant) increased the self-renewal potential and tumorigenicity of GSCs, thus mimicking the GSC phenotype in SMURF2 silencing. Mechanistically, SMURF2[Thr249] phosphorylation activates its E3 ubiquitin ligase activity, which decreases the protein stability of TGFBR via proteasome-mediated degradation. Finally, TGFBR1 silencing rescues the increased self-renewal potential and tumorigenicity of GSCs by inactivating SMURF2[Thr249] phosphorylation. Collectively, these findings highlight the importance of SMURF2[Thr249] phosphorylation in maintaining the stemness and tumorigenicity of GSCs; these findings also indicate that SMURF2[Thr249] phosphorylation could be an important posttranslational modification in treatment strategies aimed at disrupting GSCs.

## Results

**Targeting *SMURF2* promotes the self-renewal potential, invasiveness, and tumorigenicity of GSCs.** We first elucidated the functional significance of SMURF2 in maintaining GSCs in vitro by targeting *SMURF2* expression using lentiviral shRNA (sh*SMURF2*) in TGS-01 and TGS-04 GSCs, which are human GBM patient-derived GSCs. Disruption of *SMURF2* with shRNA significantly increased GSC tumorsphere formation in both TGS-01 and TGS-04 GSCs (Fig. 1a). In addition, an in vitro limiting dilution assay demonstrated that the self-renewal potential of GSCs was significantly increased by *SMURF2* silencing in both TGS-01 and TGS-04 GSCs (Fig. 1b). Furthermore, *SMURF2* knockdown resulted in significant upregulation of the stem cell markers SOX2, SOX4, NESTIN and leukemia inhibitory factor (LIF) in both TGS-01 and TGS-04 GSCs, along with a marked reduction in the SMURF2 protein level (Fig. 1c). In addition, disrupting *SMURF2* significantly increased the invasive potential (Fig. 1d); however, it did not significantly alter cell apoptosis in both TGS-01 and TGS-04 GSCs (Fig. 1e).

We next examined whether *SMURF2* silencing could affect the tumorigenic potential of GSCs in an orthotopic xenograft mouse model. Equal numbers of TGS-01 GSCs transduced with either sh*SMURF2* or sh*Control* were intracranially injected into immunocompromised mice. The mice inoculated with the sh*SMURF2*-infected TGS-01 GSCs had a significantly shortened survival compared with the mice injected with the sh*Control*-infected cells (Fig. 1f). Moreover, the histological examination demonstrated that the mice inoculated with sh*SMURF2*-infected TGS-01 GSCs displayed larger tumors compared with the mice injected with sh*Control*-infected cells (Fig. 1g). Collectively, our findings in patient-derived GSCs in vitro and in the in vivo orthotopic xenograft model indicate the importance of SMURF2 in the self-renewal potential, invasiveness, and tumorigenicity of GSCs.

**SMURF2[Thr249] phosphorylation level is lower in human GBM tissues and human GBM patient-derived GSCs.** We next assessed whether our findings were relevant to clinical data in glioma patients using publicly available datasets and our clinical samples. No marked alterations of *SMURF2* mRNA levels were found among grades II, III, and IV cancer or among classical, mesenchymal, and proneural tumors, according to the Cancer Genome Atlas (TCGA) (Fig. 2a). Moreover, in accordance with the lack of marked alterations of the *SMURF2* mRNA levels in glioma specimens in the TCGA database, we confirmed that the SMURF2 protein level was comparable between control nonneoplastic brain tissue (NB), diffuse astrocytoma (grade II), anaplastic astrocytoma (grade III), and GBM (grade IV) in our clinical samples (Fig. 2b, c).

These results led us to investigate whether the posttranslational modification of SMURF2 could be modified in human glioma specimens to reveal the functional importance of SMURF2 in the development and progression of gliomas. Given that our previous

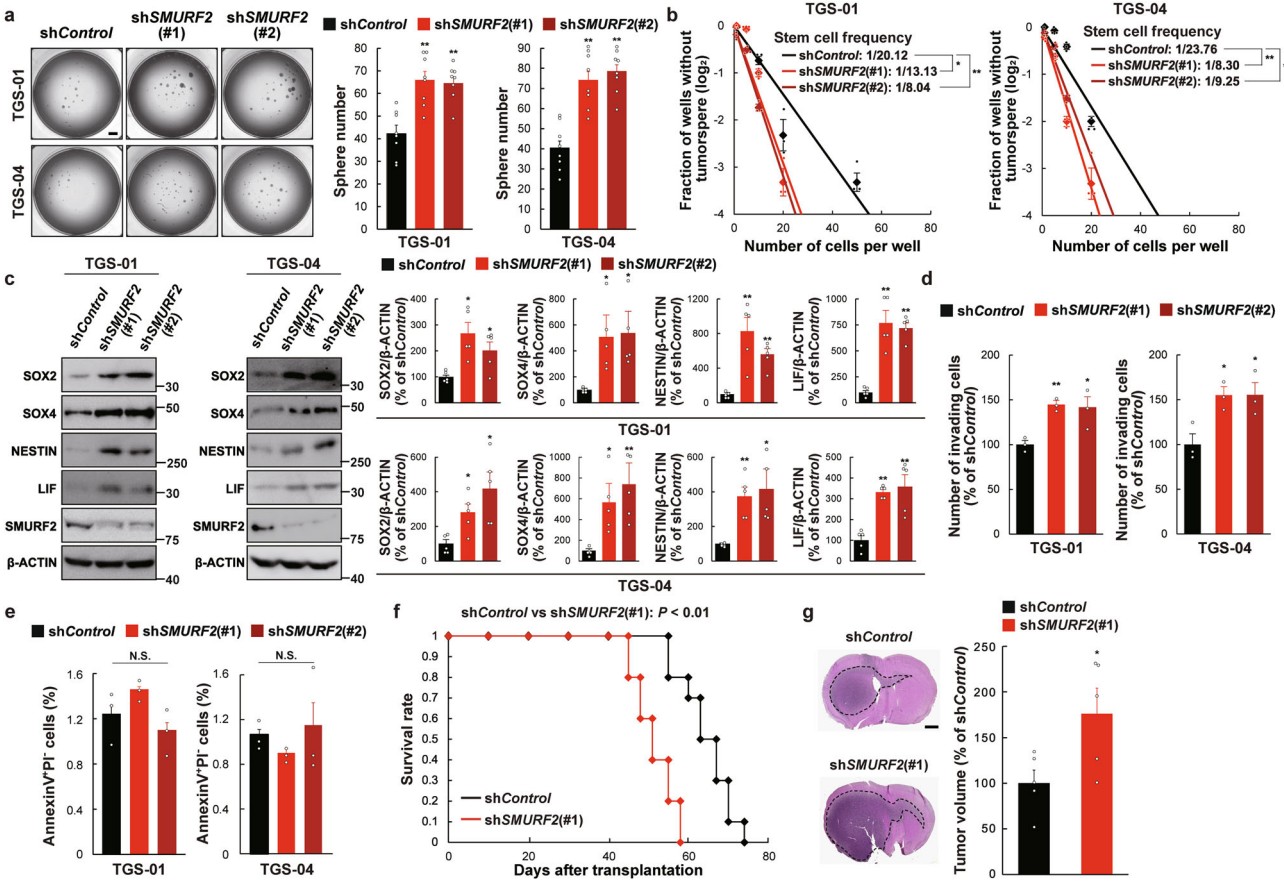

**Fig. 1 SMURF2 silencing promotes tumor growth, invasiveness, and self-renewal of GSCs.** TGS-01 and TGS-04 GSCs were infected with shSMURF2 (#1 and #2), followed by determination of (**a**) tumorsphere number ($n = 8$), (**b**) stem cell frequency by in vitro limiting dilution assay ($n = 3$) (estimated frequencies of clonogenic cells in GSC tumorsphere were calculated by ELDA analysis), (**c**) protein levels of SOX2, SOX4, NESTIN, LIF, and SMURF2; β-ACTIN served as a loading control ($n = 5$), (**d**) invasive ability by transwell assay ($n = 3$), and (**e**) cell apoptosis ($n = 3$). **f** Development of gliomas after intracranial transplantation of shSMURF2-infected TGS-01 GSCs. Survival of mice was evaluated by Kaplan–Meier analysis ($n = 10$). P value was calculated using a log-rank test. **g** Histological analyses of brains dissected at 30 days after intracranial transplantation. Tissue sections were stained with H&E ($n = 5$). *$P < 0.05$, **$P < 0.01$, significantly different from the values obtained in cells infected with shControl. N.S., not significant. Values are expressed as the mean ± S.E. and statistical significance was determined using (**a**, **c**, **d**, **e**) one-way ANOVA post hoc Bonferroni test and (**g**) Student's t test. Scale bar: 1 mm.

study reported that SMURF2[Thr249] phosphorylation plays an essential role in maintaining the stemness of MSCs[10], we next examined the SMURF2[Thr249] phosphorylation level in human glioma specimens. The SMURF2[Thr249] phosphorylation level was significantly lower in the GBM (grade IV) and anaplastic astrocytoma (grade III) specimens than in NB specimens (Fig. 2b and 2d). Moreover, the SMURF2[Thr249] phosphorylation level was negatively correlated with the protein level of SOX2, a stem cell transcription factor[31,32], in glioma specimens (Fig. 2e).

Further, we compared the SMURF2[Thr249] phosphorylation level in GSCs and that in differentiated glioma cells. For this, TGS-01 and TGS-04 cells were cultured in neurosphere culture condition (for GSCs) or adherent culture condition (for differentiated glioma cells). Under neurosphere culture condition, TGS-01 and TGS-04 GSCs displayed a significant lower SMURF2[Thr249] phosphorylation level, in addition to a higher SOX2 level and a lower GFAP level, when compared with TGS-01 and TGS-04 cells cultured under adherent culture condition (Fig. 2f). Conversely, SMURF2 protein level was comparable between cells under the two culture conditions (Fig. 2f). Therefore, these results indicated that the SMURF2[Thr249] phosphorylation level was significantly lower in GSCs than that in differentiated glioma cells. Our experimental findings aligned

with publicly available clinical data suggest that SMURF2[Thr249] phosphorylation rather than SMURF2 levels (protein and mRNA) might be associated with tumor grade and glioma stemness in humans. Thus, SMURF2[Thr249] phosphorylation may serve as a prognostic marker of GBM.

**SMURF2[Thr249] phosphorylation is implicated in the self-renewal potential, invasiveness, and tumorigenicity of GSCs.** We next determined whether the SMURF2[Thr249] phosphorylation is implicated in the maintenance of GSCs in vitro. To this end, a T249A SMURF2 mutant construct (hereafter referred to as SMURF2[T249A]), in which threonine was replaced by alanine to prevent phosphorylation, was lentivirally infected in both TGS-01 and TGS-04 GSCs. The introduction of SMURF2[T249A] significantly increased tumorsphere formation and the self-renewal potential in both TGS-01 and TGS-04 GSCs; conversely, these changes were significantly decreased after the introduction of wild-type SMURF2 (hereafter referred to as SMURF2[WT]) (Fig. 3a, b). In addition, an immunoblotting analysis revealed that the protein levels of SOX2, SOX4, NESTIN, and LIF were significantly upregulated by SMURF2[T249A] but significantly downregulated by SMURF2[WT] in both TGS-01 and TGS-04 GSCs (Fig. 3c). In addition, the invasive potential was significantly increased by SMURF2[T249A] but

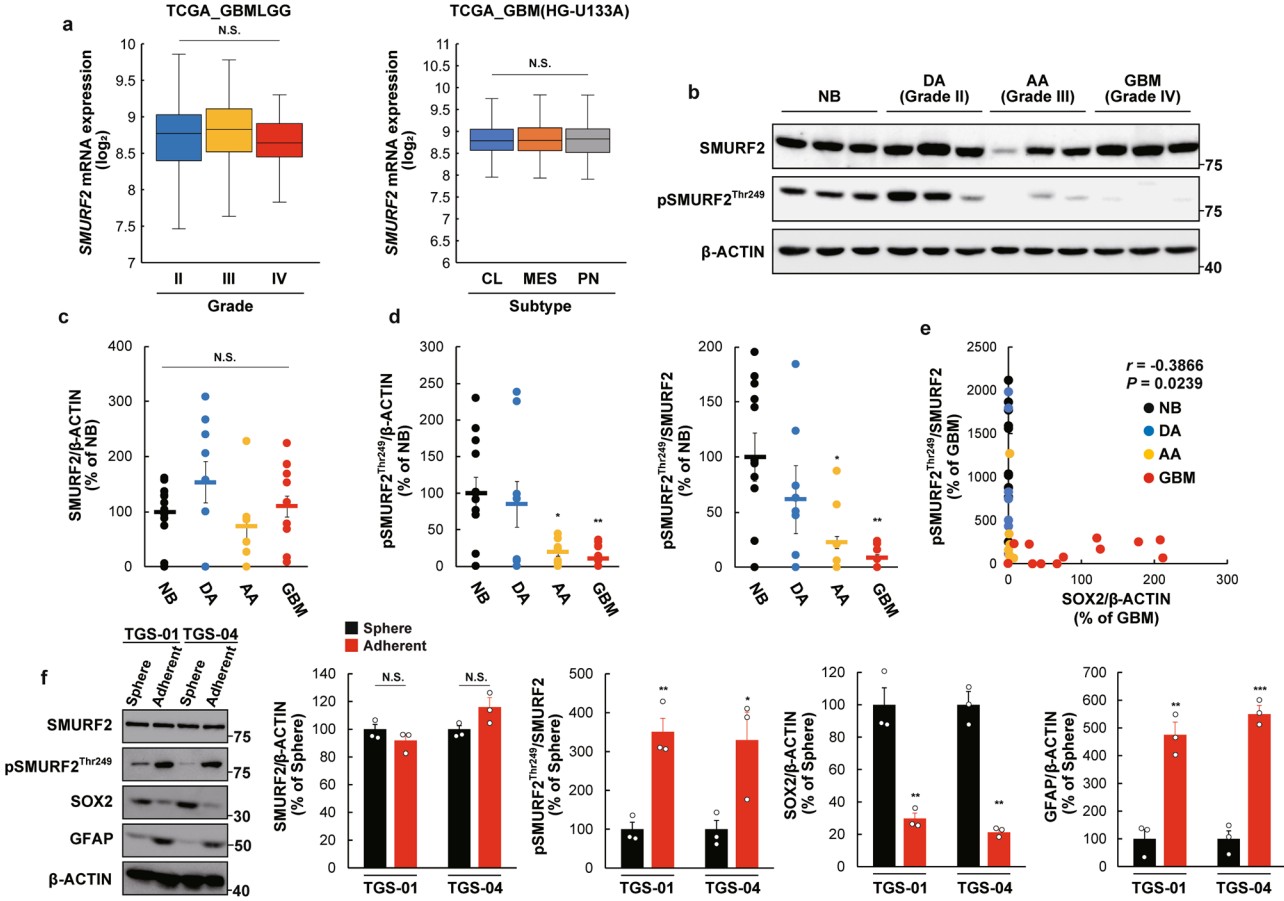

**Fig. 2 SMURF2^Thr249 phosphorylation is decreased in anaplastic astrocytoma and GBM specimens, and is a negative correlation with stem cell marker. a** mRNA expression of *SMURF2* in each grade (grade II, $n = 226$; grade III, $n = 244$; grade IV, $n = 150$) or subtype (classical (CL), $n = 199$; mesenchymal (MES), $n = 166$; proneural (PN), $n = 163$) of glioma. The data were obtained and analyzed using GlioVis database. **b–d** Determination of protein levels of SMURF2 and pSMURF2^Thr249 in human glioma samples. Nonneoplastic brain tissue (NB) ($n = 12$), diffuse astrocytoma (DA) Grade II ($n = 9$), anaplastic astrocytoma (AA) Grade III ($n = 9$), glioblastoma (GBM) Grade IV ($n = 16$). **e** Correlation between SOX2 and pSMURF2^Thr249 in glioma samples. **f** TGS-01 and TGS-04 cells were cultured in neurosphere medium or adherent culture medium, followed by determination of protein levels of SMURF2, pSMURF2^Thr249, SOX2 and GFAP; β-ACTIN served as a loading control ($n = 3$). *$P < 0.05$, **$P < 0.01$, ***$P < 0.001$, significantly different from the value obtained in (**c, d**) NB or (**f**) Sphere. N.S., not significant. Values are expressed as the mean ± S.E. and statistical significance was determined using (**a**) Tukey's Honest Significant Difference test, (**c, d**) one-way ANOVA *post hoc* Bonferroni test, and (**f**) Student's *t* test. *r*, Pearson's correlation coefficient.

significantly decreased by *SMURF2*^WT, whereas cell apoptosis was not markedly altered by either *SMURF2*^T249A or *SMURF2*^WT in both TGS-01 and TGS-04 GSCs (Fig. 3d, e).

Given that SMURF2^Thr249 phosphorylation level was higher in differentiated glioma cells than in GSCs (Fig. 2f), as well as in NB than in GBM specimens (Fig. 2b, d), we tested whether SMURF2^Thr249 phosphorylation could be implicated in the function of differentiated glioma cells in addition to the function of GSCs. Cell proliferation was not significantly altered by either *SMURF2*^T249A or *SMURF2*^WT in human differentiated glioma cells lines U87 and SNB19 (Supplementary Fig. 1a). In contrast, the invasive potential was significantly increased by *SMURF2*^T249A, but significantly decreased by *SMURF2*^WT in U87 and SNB19 cells (Supplementary Fig. 1b), as observed in GSCs, which indicated that SMURF2^Thr249 phosphorylation might contribute to the cellular function of both GSCs and differentiated glioma cells.

We next examined the impact of SMURF2^Thr249 phosphorylation on the tumorigenic potential of GSCs in vivo. Equal numbers of TGS-01 GSCs transduced with either *SMURF2*^T249A or *SMURF2*^WT were intracranially injected into immunocompromised mice. The inoculation of *SMURF2*^T249A-infected cells

significantly shortened the survival of the mice compared with the inoculation of empty vector (E.V.) -infected cells; conversely, their survival was significantly prolonged by the inoculation of *SMURF2*^WT-infected cells (Fig. 3f). Moreover, *SMURF2*^T249A-infected cells generated larger tumors than the control cells, whereas *SMURF2*^WT-infected cells generated smaller tumors (Fig. 3g). Immunoblotting analysis showed that the SOX2 level was significantly increased in the ipsilateral side than that in the contralateral side after inoculation of E.V.-infected cells and *SMURF2*^T249A-infected cells (Fig. 3h). The SOX2 level in the ipsilateral side was significantly decreased in mice inoculated with *SMURF2*^WT-infected cells than that in mice with E.V.-infected cells, whereas it tended to increase in mice inoculated with *SMURF2*^T249A-infected cells (Fig. 3h). Collectively, these results indicate that SMURF2^Thr249 phosphorylation could regulate the self-renewal potential, invasiveness, and tumorigenicity of GSCs.

**SMURF2^Thr249 phosphorylation modifies the TGF-β-SMAD2/3 axis by controlling TGFBR stability in GSCs.** The self-renewal potential and tumorigenicity of GSCs were activated by inactivating SMURF2^Thr249 phosphorylation, thus recapitulating GSC phenotypes by *SMURF2* silencing. The phosphorylation of

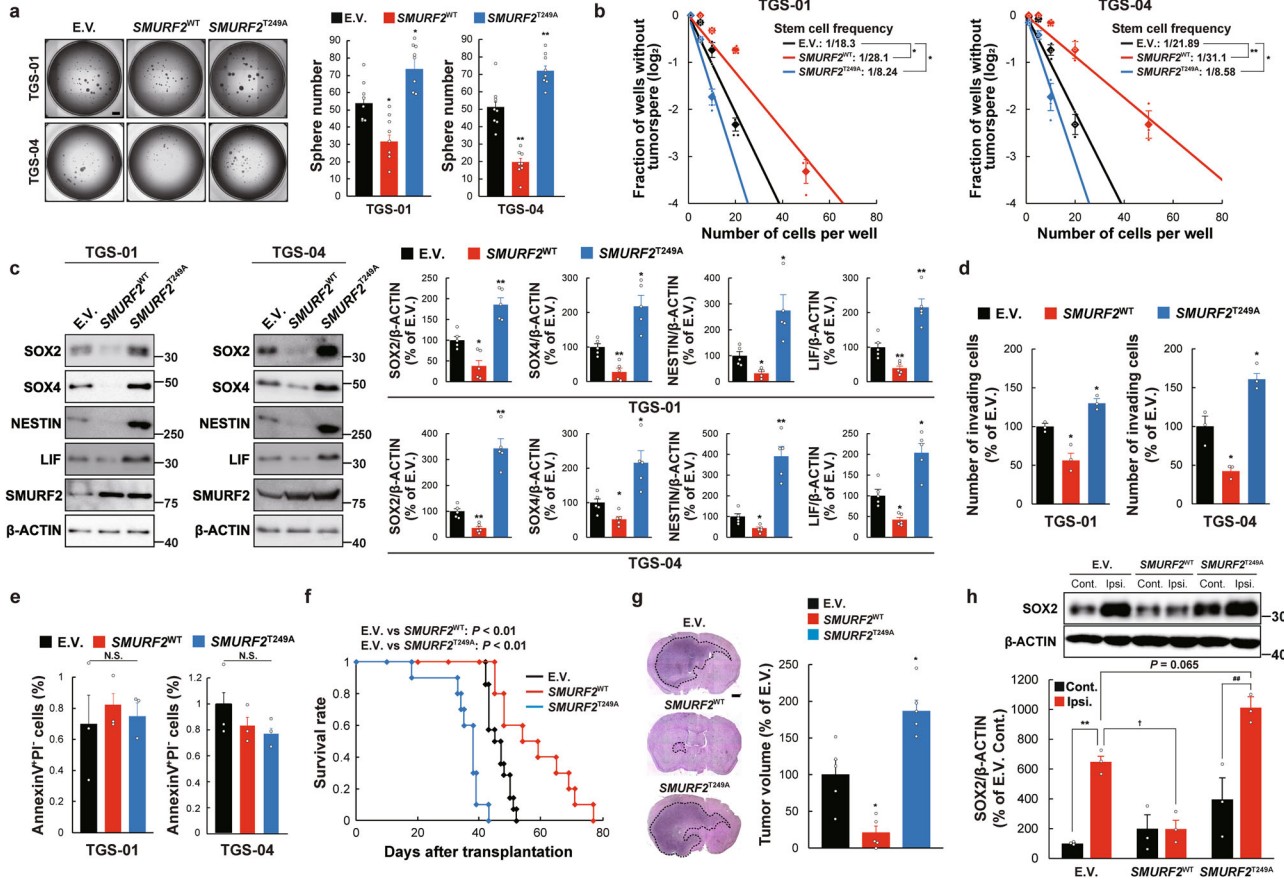

**Fig. 3 SMURF2$^{Thr249}$ phosphorylation regulates tumor growth, invasiveness, and self-renewal of GSCs.** TGS-01 and TGS-04 GSCs were infected with *SMURF2*$^{WT}$ or *SMURF2*$^{T249A}$, followed by determination of (**a**) tumorsphere number ($n = 8$), (**b**) stem cell frequency by in vitro limiting dilution assay ($n = 3$), (**c**) protein levels of SOX2, SOX4, NESTIN, LIF, and SMURF2 ($n = 5$), (**d**) invasive ability by transwell assay ($n = 3$), and (**e**) cell apoptosis ($n = 3$). **f** Development of gliomas after intracranial transplantation of *SMURF2*$^{WT}$- or *SMURF2*$^{T249A}$-infected TGS-01 GSCs. Survival of mice was evaluated by Kaplan–Meier analysis ($n = 14$). $P$ value was calculated using a log-rank test. **g** Histological analyses of brains dissected at 30 days after intracranial transplantation. Tissue sections were stained with H&E ($n = 5$). *$P < 0.05$, **$P < 0.01$, significantly different from the value obtained in cells infected with E.V. **h** Determination of protein levels of SOX2 in the brain of ipsilateral (Ipsi.) side of inoculation and contralateral (Cont.) side at 40 days after intracranial transplantation; β-ACTIN served as a loading control ($n = 3$). **$P < 0.01$, significantly different from the value obtained in Cont. side inoculated E.V.-infected cells. ##$P < 0.01$, significantly different from the value obtained in Cont. side inoculated *SMURF2*$^{T249A}$-infected cells. †$P < 0.05$, significantly different from the value obtained in Ipsi. side inoculated E.V.-infected cells. N.S., not significant. Values are expressed as the mean ± S.E. and statistical significance was determined using (**a**, **c**, **d**, **e**, **g**) one-way ANOVA *post hoc* Bonferroni test and (**h**) two-way ANOVA *post hoc* Tukey–Kramer test. Scale bar: 1 mm.

SMURF2$^{Thr249}$ activates its ubiquitin E3 ligase ability to accelerate the proteasomal degradation of SMAD proteins (SMAD1, SMAD2, and SMAD3) in MSCs to control the stemness; furthermore, the TGF-β/SMAD and BMP/SMAD axes play a crucial role in regulating the stemness and tumorigenicity of GSCs through the SMAD pathway[25,33,34]. We therefore investigated whether SMURF2$^{Thr249}$ phosphorylation could regulate the TGF-β/SMAD and BMP/SMAD axes in GSCs. The protein levels of TGFBR1 and TGFBR2 and the phosphorylation level of SMAD2 and SMAD3 were significantly increased by *SMURF2*$^{T249A}$; however, these levels were decreased by *SMURF2*$^{WT}$ in TGS-01 GSCs (Fig. 4a). Conversely, the protein levels of BMPR2 and BMPR1A and the phosphorylation level of SMAD1/5/9 were not significantly altered by either *SMURF2*$^{T249A}$ or *SMURF2*$^{WT}$ in TGS-01 GSCs (Fig. 4a). These results indicate that SMURF2$^{Thr249}$ phosphorylation may regulate the TGF-β-SMAD2/3 axis rather than the BMP-SMAD1/5/9 axis in GSCs.

To elucidate whether SMURF2$^{Thr249}$ phosphorylation controls TGFBR protein stability through the ubiquitin-proteasome pathway in GSCs, TGS-01 GSCs were treated with cycloheximide

(CHX), a protein synthesis inhibitor, and the TGFBR1 and TGFBR2 protein levels were evaluated. The TGFBR1 and TGFBR2 protein levels gradually decreased and became almost undetectable within 8 h of CHX treatment in E.V.-infected TGS-01 GSCs (Fig. 4b). The enforced expression of *SMURF2*$^{T249A}$ prominently increased the stability of both the TGFBR1 and TGFBR2 proteins whereas the introduction of *SMURF2*$^{WT}$ destabilized their proteins in TGS-01 GSCs (Fig. 4b). We next investigated the role of SMURF2$^{Thr249}$ phosphorylation in SMURF2-dependent TGFBR protein degradation. Firstly, an immunoprecipitation assay revealed that SMURF2 physically interacts with TGFBR1 and TGFBR2 in TGS-01 GSCs (Fig. 4c). Moreover, ubiquitination of endogenous TGFBR1 and TGFBR2 was markedly elevated by the overexpression of *SMURF2*$^{WT}$, but it was decreased by the enforced infection of *SMURF2*$^{T249A}$ in TGS-01 GSCs (Fig. 4d). These results suggest that SMURF2$^{Thr249}$ phosphorylation decreases the protein stability of TGFBR by enhancing its E3 ubiquitin ligase activity, which in turn reduced TGF-β-SMAD2/3 signaling to repress the self-renewal potential and tumorigenicity of GSCs.

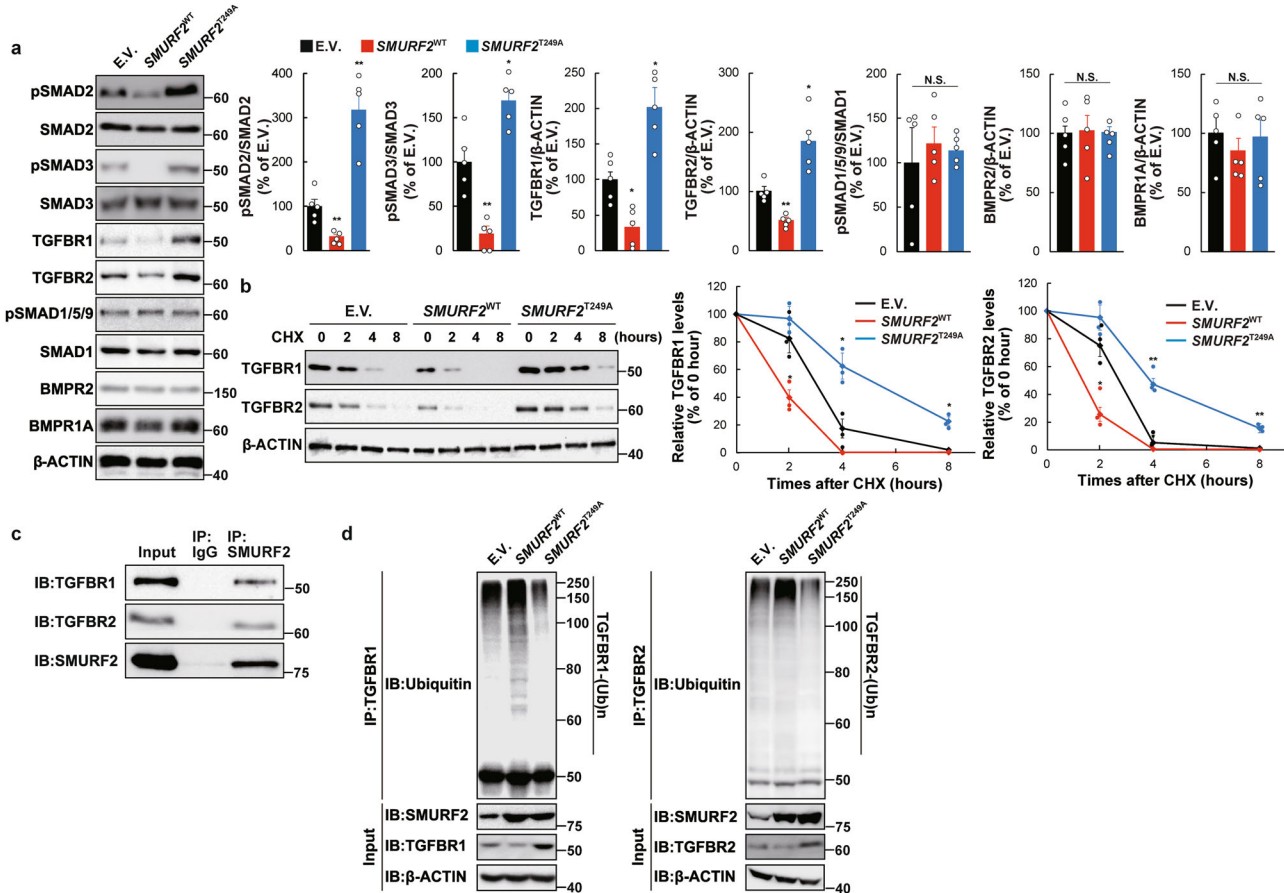

**Fig. 4 SMURF2^Thr249 phosphorylation regulates ubiquitin-dependent degradation of TGFBR protein. a** TGS-01 GSCs were infected with *SMURF2*^WT or *SMURF2*^T249A, followed by determination of protein levels by immunoblotting (*n* = 5). **b** TGS-01 GSCs were infected with *SMURF2*^WT or *SMURF2*^T249A, and treated with cycloheximide (CHX) at 50 μg/ml for indicated hours, followed by immunoblotting (*n* = 3). **c** Immunoprecipitation assay was performed in TGS-01 GSCs (*n* = 3). **d** TGS-01 GSCs were infected with *SMURF2*^WT or *SMURF2*^T249A, and subsequent immunoprecipitation with anti-TGFBR1 antibody or anti-TGFBR2 antibody, followed by determination of Ubiquitin with anti-Ubiquitin antibody (*n* = 3). *$P < 0.05$, **$P < 0.01$, significantly different from the value obtained in cells infected with E.V., N.S., not significant. Values are expressed as the mean ± S.E. and statistical significance was determined using (**a**) one-way ANOVA *post hoc* Bonferroni test, and (**b**) two-way ANOVA *post hoc* Bonferroni test.

**TGFBR1 is a critical target through which SMURF2^Thr249 phosphorylation can regulate the self-renewal potential and tumorigenicity of GSCs.** We next examined whether the activation of TGF-β signaling by TGFBR protein stability could contribute to the regulation of self-renewal potential and tumorigenicity of GSCs by SMURF2^Thr249 phosphorylation. *TGFBR1* silencing in TGS-01 GSCs significantly attenuated the increased tumorsphere formation and self-renewal potential caused by the introduction of *SMURF2*^T249A (Fig. 5a, b). In addition, *TGFBR1* knockdown significantly rescued the shortened duration of survival in mice bearing *SMURF2*^T249A-infected TGS-01 GSCs, resulting in an increased rate of prolonged survival (Fig. 5c, d). Finally, SMURF2^Thr249 phosphorylation was negatively correlated with the TGFBR1 protein levels in human glioma specimens (Fig. 5e). These results indicate that the phosphorylation of SMURF2^Thr249 is important for regulating TGFBR1 protein stability to control the self-renewal potential and tumorigenicity of GSCs.

## Discussion

The E3 ubiquitin ligase activity of SMURF2 is regulated at the posttranslational level, including through phosphorylation[8,35,36]. SMURF2 activity is inhibited by the phosphorylation at Tyr^314/Tyr^434 by c-Src and Ser^384 by ATM[35,36]. We recently demonstrated that SMURF2^Thr249 phosphorylation by ERK5 activates its

ubiquitin ligase activity and subsequently controls the stemness of MSCs through modulating the SMAD-SOX9 molecular axis, thus contributing to skeletogenesis[10]. In this study, SMURF2^Thr249 phosphorylation controlled the stemness, invasiveness, and tumorigenicity of GSCs by modulating the TGFBR-SMAD-SOX4 molecular axis, contributing to gliomagenesis (Fig. 5f), and downregulating SMURF2^Thr249 phosphorylation in human GBM tissues as well as human GBM patient-derived GSCs. Given that the high expression levels of endogenous ERK5 are associated with higher grade tumors and poorer survival in adult gliomas, ERK5 could unlikely be implicated in the regulation of SMURF2^Thr249 phosphorylation in gliomas[37]. Although candidate kinases and/or phosphatases responsible for controlling SMURF2^Thr249 phosphorylation could be predicted based on in silico approaches (NetPhos and DEPOD), further studies should be performed to identify key factors that have an important role in GSCs. In any case, our results demonstrated that SMURF2^Thr249 phosphorylation may be a crucial posttranslational modification for modulating the stemness and tumorigenicity of GSCs, thereby suggesting that molecules that modify the activities of kinases and/or phosphatases responsible for SMURF2^Thr249 phosphorylation could be novel potential GSC-targeting drugs.

SMURF2 is considered to perform a dual role as a promoter and suppressor of tumors by regulating the stability of certain

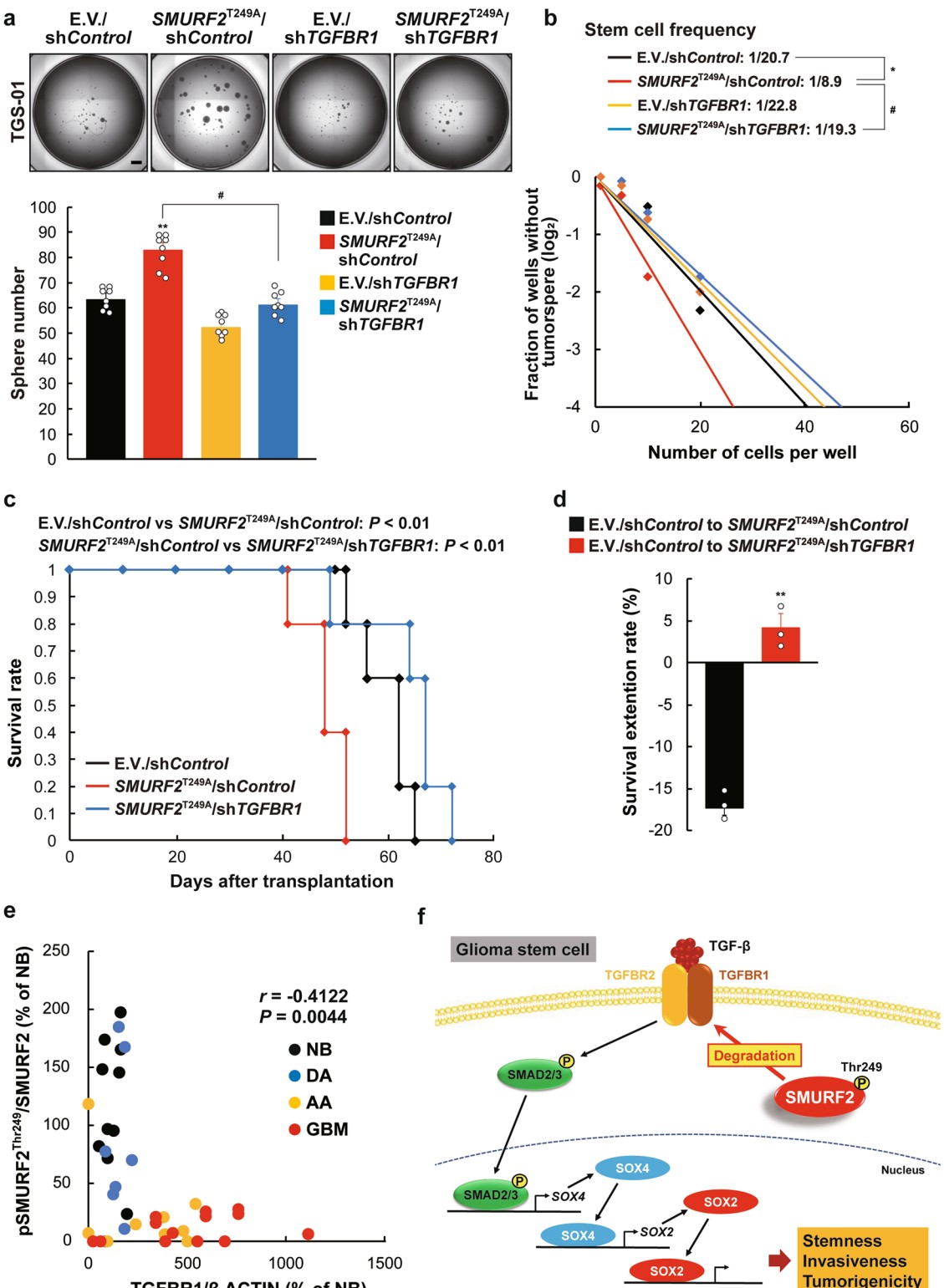

**Fig. 5 *TGFBR1* silencing restores the promotive effect of *SMURF2*[T249A] on GSC phenotypes.** TGS-01 GSCs were infected with *SMURF2*[T249A] and/or sh*TGFBR1*, followed by determination of (**a**) tumorsphere number ($n = 8$), (**b**) stem cell frequency by in vitro limiting dilution assay. **c** Development of gliomas after intracranial transplantation of *SMURF2*[T249A]- and/or sh*TGFBR1*-infected TGS-01 GSCs. Survival of mice was evaluated by Kaplan–Meier analysis ($n = 5$). *P* value was calculated using a log-rank test. **d** Survival extension rate. **e** Correlation between TGFBR1 and pSMURF2[Thr249] in glioma samples shown in Fig. 2. **f** Schematic model of the findings of this study. Phosphorylation of SMURF2[Thr249] enhances ubiquitin-dependent degradation of TGFBR protein, which results in the repression of SMAD2/3-SOX4/2 axis, leading to the inhibition of stemness, invasiveness, and tumorigenicity of GSCs. (a and b) \**P* < 0.05, \*\**P* < 0.01, significantly different from the value obtained in cells infected with E.V./sh*Control*. #*P* < 0.05, significantly different from the value obtained in cells infected with *SMURF2*[T249A]/sh*Control*. Values are expressed as the mean ± S.E. and statistical significance was determined using (**a**) two-way ANOVA *post hoc* Bonferroni test and (**d**) Student's *t* test. *r*, Pearson's correlation coefficient. Scale bar: 1 mm.

proteins involved in tumorigenesis in cell-dependent and context-dependent manners. SMURF2 interacts with and destabilizes H2AX, which plays a central role in DNA repair and genome stability, in glioma cells[38]. SMURF2 silencing reduces the migration and invasion of breast carcinomas and colorectal cancer[3,39]. Moreover, SMURF2 is overexpressed in some types of ovarian cancers and breast cancers[4], and high expression levels of SMURF2 are related to poor prognosis in esophageal carcinomas[40], suggesting that SMURF2 acts as a tumor promoter in certain tumors. Conversely, mouse genetic studies have revealed that SMURF2 deficiency leads to an increase in the possibility of a wide spectrum of tumors in various tissues and organs including the liver, blood, and lungs in aged mice[41], thus implicating SMURF2 as a potent tumor suppressor. However, the mechanisms underlying SMURF2 activity in human malignancies remain elusive because SMURF2 is rarely found mutated or deleted in cancers[42]. Here, we show that the disruption of SMURF2 resulted in an enhancement of the self-renewal potential and tumorigenicity of GSCs, which are phenocopied by inactivation of SMURF2 by a non-phosphorylatable mutant; conversely, the opposite reaction was observed through SMURF2 overexpression in GSCs. Moreover, SMURF2$^{Thr249}$ phosphorylation was markedly lower in the GBM pathology specimens, accompanied by no marked alteration in the SMURF2 protein level, irrespective of the unknown mechanism of downregulated SMURF2$^{Thr249}$ phosphorylation in GBM patients. Although we should investigate whether SMURF2$^{Thr249}$ phosphorylation has a prognostic value for glioma patients, SMURF2 could exert tumor suppressor functions in glioma pathogenesis, in which SMURF2 activity is controlled by SMURF2$^{Thr249}$ phosphorylation status rather than SMURF2 expression levels.

The functional role of SMURF2 on tumorigenesis has been reported to be connected to its ability to regulate the protein stability of a variety of substrate repertories, in addition to altering the cellular distribution of SMURF2[3,41]. For example, SMURF2 governs the chromatin organization, dynamics, and genome integrity by controlling the proteasomal degradation or the protein stability of its substrates including RNF20 or DNA topoisomerase IIa[41,43], which in turn regulate tumorigenesis and tumor progression. Moreover, SMURF2 regulates the stability of pro-oncogenic transcription factors such as KLF5, YY1, and ID1[44–46], in addition to regulating Wnt/β-catenin oncogenic signaling and KRAS oncoproteins[47–49]. Although we show here that SMURF2$^{Thr249}$ phosphorylation plays a crucial role in stemness and tumorigenicity by modulating TGF-β signaling through the ubiquitin-proteasome-dependent degradation of TGFBR proteins in GSCs, it should be emphasized that additional molecular mechanisms might be involved in the control of tumorigenicity in GSCs by SMURF2$^{Thr249}$ phosphorylation.

In conclusion, SMURF2$^{Thr249}$ phosphorylation plays a crucial role in glioma pathogenesis by modulating TGF-β/SMAD signaling in GSCs. To our knowledge, this is the first preclinical study to investigate the functional role of SMURF2 on the function of cancer stem cells in vivo. Our findings improve our understanding of the molecular mechanism underlying the maintenance of the stemness and tumorigenicity of GSCs and suggest that SMURF2$^{Thr249}$ phosphorylation status could represent a novel target for drug development to treat not only gliomas but also malignant tumors associated with the aberrant expression or function of TGF-β signaling in humans.

## Materials and methods

**Cell culture and reagents**. HEK293T cells were purchased from RIKEN BRC (#RCB2202), and were cultured in Dulbecco's modified Eagle's medium (DMEM) (FUJIFILM Wako Pure Chemical #043-30085) supplemented with 10% fetal bovine serum (FBS). Human glioma cell lines U87 and SNB19 were purchased from ATCC (#HTB-14, #CRL-2219), and were cultured in DMEM supplemented with 10% FBS. Human patient-derived GBM cell lines TGS-01 and TGS-04 were established as described previously[25]. The use of these human materials and protocols were approved by the Ethics Committees of Gifu Pharmaceutical University, Kanazawa University, and the University of Tokyo. These cells were cultured in neurosphere medium containing DMEM/F12 (FUJIFILM Wako Pure Chemical #048-29785) supplemented with GlutaMAX (Gibco #35050061), B27 supplement minus vitamin A (Gibco #12587010), 20 ng/ml recombinant human epidermal growth factor (FUJIFILM Wako Pure Chemical #059-07873) and 20 ng/ml recombinant human basic fibroblast growth factor (FUJIFILM Wako Pure Chemical #064-04541). These cells were differentiated in adherent culture medium containing DMEM supplemented with 10% FBS for 7 days.

**Cell proliferation assay and apoptosis assay**. Cell suspensions were prepared using 0.25% Trypsin and 1 mM EDTA solution, filtered through a 70 μm cell strainer (BD Biosciences), and counted using a TC20 automated cell counter (Bio-Rad). Cells were dissociated into single cells with StemPro Accutase (Gibco). Apoptosis assay was conducted using the FITC-Annexin V Apoptosis Detection kit (BD Biosciences, #556547) and analyzed by BD FACS Verse and BD FACSuite software. The gating strategies for apoptotic cells in the paper are available in Supplementary Fig. 2.

**Transwell invasion assay**. Cell suspensions were prepared using 0.25% Trypsin and 1 mM EDTA solution and filtered through a 70 μm cell strainer (BD Biosciences). Cells were seeded onto the upper chamber of Transwell with 8 μm pore membrane filters (Corning #3422) in a serum-free DMEM. DMEM containing 10% FBS was added to the lower chamber. After 24 h of incubation, the cells that had invaded the lower surface of the filter were fixed with 4% paraformaldehyde/PBS and stained with 0.1% crystal violet. Invasive cells were captured using a BZ-X810 fluorescence microscope (Keyence) and analyzed for quantitating cell numbers using BZ-X810 Analyzer software (Keyence).

**Surgical specimens**. A total of 46 primary glioma tissues were obtained from patients who underwent surgical removal of tumor. The specimens were reviewed and classified according to WHO criteria[50]. Nonneoplastic healthy brain tissues adjacent to tumors were acquired. The tissues were homogenized in lysis buffer. All experiments were approved by the local Institutional Review Board of Kanazawa University (No. 2509) and all study participants provided written informed consent.

**Immunoblotting analysis**. Cells were solubilized in lysis buffer (10 mM Tris-HCl, 150 mM NaCl, 0.5 mM EDTA, 10 mM NaF, 1% Nonidet P-40, pH 7.4) containing protease inhibitor cocktail. Samples were then subjected to SDS-PAGE, followed by transfer to polyvinylidene difluoride (PVDF) membranes and subsequent immunoblotting. The primary antibodies used were, anti-p-Smurf2$^{Thr249}$ (#J1683BA260-5, 1:2000) (GenScript), anti-Phospho-Smad2 (Ser465/467) (#3101, 1:1000), anti-Smad2 (#5339, 1:1000), anti-Phospho-Smad3 (Ser423/425) (#9520, 1:1000), anti-Smad3 (#9523, 1:1000), anti-TGFBR I (#3712, 1:1000), anti- TGFBR II (#79424, 1:1000), anti-Sox2 (#3579, 1:1000), anti-p-Smad1 (Ser463/465)/5 (Ser463/465)/9 (Ser465/467) (#13820, 1:1000), anti-Smad1 (#9512, 1:1000), anti-BMPR2 (#6979, 1:1000) and anti-Ubiquitin (#3936, 1:1000) (Cell Signaling Technologies), anti-β-actin (#sc-47778, 1:2000) (Santa Cruz Biotechnology), anti-Sox4 (#AB5803, 1:1000) (EMD Millipore), anti-NESTIN (#ab105389, 1:1000), anti-LIF (#ab138002, 1:1000), anti-BMPR1A (#ab174815, 1:1000), and anti-SMURF2 (#ab94483, 1:1000) (Abcam). The primary antibodies were diluted with blocking solution (5% skim milk). The custom polyclonal p-Smurf2$^{Thr249}$ antibody was generated (#J1683BA260-5) (GenScript). Briefly, two rabbits were injected with KLH-conjugated p-Smurf2$^{Thr249}$ epitope, representing amino acids 244-258, emulsified in Freund's complete adjuvant, and then boosted three times at 14-day intervals with p-Smurf2$^{Thr249}$ epitope. The images were acquired using ChemiDoc Touch Imaging System (Bio-Rad). Quantification was performed by densitometry using ImageJ.

**Tumorsphere formation assay and in vitro limiting dilution assay**. For sphere formation assay, single cell suspensions were prepared using StemPro Accutase (Gibco, #A1110501) and filtered through a 70 μm cell strainer (BD Biosciences). Cells were then plated in 96-well Costar ultra-low attachment plate (Corning) at $2 \times 10^3$ cells per well with neurosphere medium mixed with 1% methylcellulose. Tumorsphere number were measured on day 7. For in vitro limiting dilution assay, cells were plated in 96-well plate at 1, 5, 10, 20 or 50 cells per well, with 10 replicates for each cell number. The presence of tumorspheres in each well was examined on day 7. Cell images were captured using a BZ-X810 fluorescence microscope (Keyence) and analyzed for quantitating sphere numbers and sizes using BZ-X810 Analyzer software (Keyence). Limiting dilution assay analysis was performed using online software (http://bioinf.wehi.edu.au/software/elda/). Sphere formation was estimated by scoring the number of spheres larger than 50 μm.

**Orthotopic xenograft model of GSC-derived GBM and histology**. Orthotopic xenograft model of GSC-derived GBM was generated by transplantation of $5 \times 10^4$ TGS-01 GSCs into the brain of 4-week-old female nude mice (BALB/cSlc-nu/nu, SLC, Shizuoka, Japan). Briefly, a small burr hole was drilled in the skull 0.5 mm anterior and 2.0 mm lateral from bregma with a micro drill, and dissociated cells were transplanted at a depth of 3 mm below the dura mater. Mice were sacrificed at the indicated time points or upon occurrence of neurological symptoms. Mouse brains were fixed with 4% paraformaldehyde solution, embedded in paraffin, and then sectioned at a thickness of 5 μm. Sections were stained with Hematoxylin and Eosin (H&E). The sections were captured using a BZ-X810 fluorescence microscope (Keyence). The tumor area was outlined and calculated using ImageJ in each section. Slide tumor volumes were calculated by multiplication of tumor area with the slice thickness 5 μm, and brain tumor volumes were approximated by adding up the slide tumor volumes for each animal. All animal experiments were approved by the Committees on Animal Experimentation of Gifu Pharmaceutical University and Kanazawa University and performed in accordance with the guidelines for the care and use of laboratory animals. The numbers of animals used per experiment are stated in the figure legends.

**Generation of lentiviral vectors and infection**. The lentiviral SMURF2 mutant vector was previously generated[10]. The oligonucleotides for SMURF2 short hairpin RNA (shRNA) were synthesized (Supplementary Table 1), annealed, and inserted into the mCherry vector, and the shRNA vector for TGFBR1 was obtained from Sigma (SHCLNG-NM_004612, TRCN0000196326). These vectors were then transfected into HEK293T cells using the calcium phosphate method. Virus supernatants were collected 48 h after transfection and cells were then infected with viral supernatant for 24 h.

**Immunoprecipitation (IP) assay**. Cells were solubilized in lysis buffer (10 mM Tris-HCl, 150 mM NaCl, 0.5 mM EDTA, 10 mM NaF, 1% Nonidet P-40, pH 7.4) containing protease inhibitor cocktail. Samples were incubated with an antibody in lysis buffer for 24 h at 4 °C and subsequent IP with protein G-Sepharose. Immunoprecipitates were washed three times with lysis buffer and boiled in SDS sample buffer. Samples were then separated by SDS-PAGE, followed by transfer to PVDF membranes and subsequent immunoblotting.

**Bioinformatics**. Gene expression data from TCGA project was obtained and analyzed using GlioVis database (http://gliovis.bioinfo.cnio.es/).

**Statistical analysis and reproducibility**. Unless otherwise specified, two-sided Student's $t$ test and one-way ANOVA post hoc Bonferroni test were used for statistical significance. For correlation analysis, we calculated Pearson's correlation coefficient. Throughout this study, $P < 0.05$ were considered statistically significant. All experiments were taken from distinct samples and the number of biological replicates ($n$) is indicated in figure legends.

**Reporting summary**. Further information on research design is available in the Nature Research Reporting Summary linked to this article.

## Data availability

The bioinformatic data that support the findings of this study are openly available in GlioVis database (http://gliovis.bioinfo.cnio.es/). (see 'Materials and Methods' section). The uncropped images of the membranes used for immunoblotting are available in Supplementary Figs. 3–11. The source data for all the plot in the paper are included in the Supplementary Data 1 file.

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

## Acknowledgements

This work was supported in part by the Japan Society for the Promotion of Science (20H03407 to E.H.); and a grant from Japan Research Foundation for Clinical Pharmacology (to E.H.).

## Author contributions

M.H., K.F., T.I., and E.H. conceived the project. M.H., K.F., T.I., T.H., K.T., S.I., M.M., M.K., A.S., and G.P. performed the experiments and analysis. H.S., T.T., A.H., and M.N. provided critical reagents. K.K., T.T., A.H., and M.N. discussed the results, conceived some experiments. M.H., K.F., T.I., and E.H. wrote the paper.

## Competing interests

The authors declare no competing interests.
