## [Transparent Peer Review File · Communications Biology]

Reviewers' comments:

Reviewer #1 (Remarks to the Author):

Hiraiwa et al describe in the article "SMURF2 phosphorylation at Thr249 modifies the stemness and tumorigenicity of glioma stem cells by regulating TGF- β receptor stability" how SMURF2 phosphorylation in Thr249 leads to SMURF2 activation of its ubiquitin ligase activity, which results in degradation of TGFBR1, inactivation of TGF-beta signaling with consequent loss of stem cell properties and tumorigenicity capacity of glioblastoma cells. This work reveals a new mechanism of regulation of the TGF β signaling pathway to control stem cell status in glioblastoma. The work is very well design and contributes to the GBM field, but some things should be address prior publication.

1. Fig 1C, D: Repeat in TGS-04.

2. Fig 3C, D: Repeat in TGS-04.

3. The authors mainly use SOX2 and SOX4 as stem cell marker in the immunoblots, they could also include other stem cell markers such as nestin, LIF (which is also downstream of TGFbeta signaling in GBM), and cleaved NOTCH1.

4. Fig 4A. It would be better to use specific antibodies for phospho-Smad2 and phospho-smad3 separately; instead of using an antibody that recognizes both. Also blot for total smad3.

5. Fig 4. C, D. SMURF2 has been shown to interact with TGFBR2 when Smad7 is present. In the GSCs cells, does TGFBR2 bind to SMURF2? if so is its ubiquitination also affected by SMURF-T49 phosphorylation?

6. Is ERK5 responsible for P-Thr249 in GBM or lower grade gliomas, as it is in MSC? If not, could the authors speculate how is Thr249 phosphorylated in GBM?

Reviewer #2 (Remarks to the Author):

General comment

This manuscript reports data demonstrating the role of SMURF2 phosphorylation in the regulation of stemness and tumorigenesis in GBM. This is a well planned study with a conclusion supported by elegant data.

Specific comments:

Requires limited grammatical corrections.

Reviewer #3 (Remarks to the Author):

The regulation of glioma stem cells (GSC) is thought to be important in GBM's profound resistance to therapy. TGF β is highly expressed in GBM and indications of its activity is associated with resistance to standard of care radiation and temozolomide. The study by Hiraiwa and colleagues reports that phosphorylation of SMURF2 at Thr249 enhances ubiquitin-dependent degradation of TGF β type 1 receptor (TGFBR1) protein. TGFBR1 signaling phosphorylates SMAD 2/3, which promotes a SMAD2/3-SOX4/2 axis that increases stemness and tumorigenicity. Thus, SMURF mediated ubiquitination leads

to decreased TGF β receptor signaling via phosphorylation of SMAD, which reduces stemness and tumorigenicity in human GBM cells.

These conclusions are well-supported by the data, which include in vivo studies and human correlates, presented in the manuscript. However, Kavsak reported in 2000 that Smad7 binds to Smurf2 to form an E3 ubiquitin ligase that targets the TGF β receptor for degradation and Eichorn reported in 2012 that USP15 stabilization of TGFBR1 promotes glioblastoma through activation of TGF- β signaling. The new information provided by Hiraiwa is that SMURF2 Thr249 phosphorylation, which they previously identified in mesenchymal stem cells, mediates ubiquitination TGFBR1, and thus degradation that impacts GSC.

SMURF2 activity is controlled by SMURF2Thr249 phosphorylation status rather than SMURF2 expression. The ratio of SMURF/SMURF2Thr249 phosphorylation was markedly lower in the GBM pathology specimens compared to lower grade gliomas, suggesting that its regulation may mediate GBM aggressiveness, which the authors suggest is due to TGF β 's role in GSC maintenance. In this regard, I noticed that the tumors depicted in Fig. 3F for the EV and SMURF2T249A are also very invasive, apparently crossing the corpus callosum, which raises the question of whether SMURF2 also affect other behaviors than stemness. Have aggressive phenotypes been assessed by invasion assays or other means?

Inhibition of TGF β signaling upon TGFBR degradation reduces, whereas receptor stabilization increases, GSC as measured by either sphere formation or tumor growth, but the authors do not demonstrate that SMURF2 function is specific to GSC. Neurosphere GSC cultures enrich for 'stemness' but are not exclusively GSC. To provide more insight into TGF β signaling in GSC, the authors could separate GSC from more differentiated GBM cells to determine if SMURF2 acts generically (which is what I suspect from the WB data) or is specific to this population.

Additional comments:

Figure 2 A can be eliminated and reported in text.

Figure 2 E is unnecessary as it simply is another representation of the data shown in 2B-D.

Figure 3 B should have error bars and indicate the number of replicates.

Figure 3 F, how many sections were measured per tumor? The volume can be misrepresented by position unless there is a fiducial (e.g. injection site).

Figure 5E legend should state whether these are specimens shown in Figure 2. It appears that there are two relationships, one for NB and DA and another for AA and GBM. I am not sure that Pearson's correlation coefficient is the correct metric given the data distribution.

Replies to the Reviewer 1:

1. Fig 1C, D: Repeat in TGS-04.

2. Fig 3C, D: Repeat in TGS-04.

3. The authors mainly use SOX2 and SOX4 as stem cell marker in the immunoblots, they could also include other stem cell markers such as nestin, LIF (which is also downstream of TGFbeta signaling in GBM), and cleaved NOTCH1.

<Reply>

We thank the reviewer for these comments. According to these comments, we have performed additional experiments. Immunoblotting revealed that protein levels of stem cell markers (SOX2, SOX4, NESTIN and LIF) were significantly increased by shSMURF2 in both TGS-01 and TGS-04 cells. On the contrary, these protein levels were significantly decreased by SMURF2^{WT} but significantly increased by SMURF2^{T249A} in both TGS-01 and TGS-04 cells. Conversely, cell apoptosis was not significantly altered by either shSMURF2, SMURF2^{WT}, SMURF2^{T249A} in both TGS-01 and TGS-04 cells. These new data have been presented as Fig. 1C, 1E, 3C and 3E in the revised manuscript. Accordingly, several sentences have been inserted into the Results section (lines 114-119 and 174-179).

4. Fig 4A. It would be better to use specific antibodies for phospho-Smad2 and phospho-smad3 separately; instead of using an antibody that recognizes both. Also blot for total smad3.

<Reply>

According to this comment, we have performed additional experiments. These new data have been presented as Fig. 4A in the revised manuscript. Accordingly, the sentence has been inserted into the Results section (lines 215-217).

5. Fig 4. C, D. SMURF2 has been shown to interact with TGFBR2 when Smaf7 is present. In the GSCs cells, does TGFBR2 bind to SMURF2? if so is its ubiquitination also affected by SMURF-T49 phosphorylation?

<Reply>

This is a critical point. According to this comment, we have performed additional experiments. Immunoprecipitation assay revealed that SMURF2 physically interacts with TGFBR2, and ubiquitination assay revealed that endogenous TGFBR2 ubiquitination was markedly increased by SMURF2^{WT}, but decreased by SMURF2^{T249A} in TGS-01 GSCs. These new data have been presented as Fig. 4C and 4D in the revised manuscript. Accordingly, several sentences have been inserted into the Results section (lines 230-237).

6. Is ERK5 responsible for P-Thr249 in GBM or lower grade gliomas, as it is in MSC? If not, could the authors speculate how is Thr249 phosphorylated in GBM?

<Reply>

We fully agree with this comment. To address this comment, several sentences have been inserted into the Discussion section (lines 262-272).

Replies to the Reviewer 2:

Requires limited grammatical corrections.

<Reply>

In line with this comment, the manuscript has been carefully reviewed by an experienced editor whose first language is English and who specializes in editing papers written by scientists whose native language is not English.

Replies to the Reviewer 3:

Have aggressive phenotypes been assessed by invasion assays or other means?

<Reply>

We thank the reviewer for this comment. To address this comment, we have performed additional experiments. Transwell assay revealed that invasive potential was significantly increased by shSMURF2 and SMURF2^{T249A}, but significantly decreased by SMURF2^{WT} in both TGS-01 and TGS-04 cells. These new data have been presented as Fig. 1D and 3D in the revised manuscript. Accordingly, several sentences have been inserted into the Results (lines 106-107, 117-119, 127-129, 165-166, 176-179, 202-204, 259-262), and Materials and Methods (lines 341-349) sections.

To provide more insight into TGFβ signaling in GSC, the authors could separate GSC from more differentiated GBM cells to determine if SMURF2 acts generically (which is what I suspect from the WB data) or is specific to this population.

<Reply>

We fully agree with this comment. To address this comment, we have performed additional experiments. Cell proliferation was not significantly altered by either SMURF2^{WT}, or SMURF2^{T249A} in human U87 differentiated GBM cells. On the contrary, invasive potential was significantly increased by SMURF2^{T249A}, but significantly decreased by SMURF2^{WT} in U87 cells, as observed in GSCs. These results suggest that SMURF2^{Thr249} phosphorylation contributes to the cellular function of differentiated glioma cells in addition to GSCs. These new data have been presented as Supplementary Fig.1A and 1B in the revised manuscript. Accordingly, several sentences have been inserted into the Results (lines 180-189), and Materials and Methods (lines 321-323 and lines 334-337) sections.

Additional comments:

Figure 2 A can be eliminated and reported in text.

Figure 2 E is unnecessary as it simply is another representation of the data shown in 2B-D.

<Reply>

We thank the reviewer for these suggestions. We would like to keep these data in main figures, because of their importance. Figure 2A supports our in vitro and in vivo data by publicly available datasets, and Figure 2E indicates the SMURF2^{Thr249} phosphorylation is negatively correlated with stem cell marker in glioma specimens.

Figure 3 B should have error bars and indicate the number of replicates.

<Reply>

According to the comment, we have added the error bars in Figure 3B and the number of replicates in Figure legend.

Figure 3 F, how many sections were measured per tumor? The volume can be misrepresented by position unless there is a fiducial (e.g. injection site).

<Reply>

This is a critical point. We have measured approximately 50-100 slides per tumor. According to this comment, several sentences have been inserted in the Material and method section (lines 402-405).

Figure 5E legend should state whether these are specimens shown in Figure 2. It appears that there are two relationships, one for NB and DA and another for AA and GBM. I am not sure that Pearson's correlation coefficient is the correct metric given the data distribution.

<Reply>

According to the comment, we have changed the sentence in the Figure legend. Because the observations are normally distributed in Figure 5E, Pearson's correlation coefficient should be the correct metric.

REVIEWERS' COMMENTS:

Reviewer #1 (Remarks to the Author):

Thanks for covering all the comments I had for your manuscript in the revised version.

Replies to the Reviewer 1:

Thanks for covering all the comments I had for your manuscript in the revised version.

We thank the reviewer for this comment.